# Colorectal Cancer Risk Prediction Using the rs4939827 Polymorphism of the *SMAD7* Gene in the Romanian Population

**DOI:** 10.3390/diagnostics14020220

**Published:** 2024-01-19

**Authors:** Lucian-Flavius Herlo, Raluca Dumache, Ciprian Duta, Octavia Vita, Adriana Marina Mercioni, Lavinia Stelea, Roxana Sirli, Stela Iurciuc

**Affiliations:** 1Doctoral School, “Victor Babes” University of Medicine and Pharmacy, 300041 Timisoara, Romania; flavius.herlo@umft.ro; 2Department of Forensic Medicine, Bioethics, Medical Ethics and Medical Law, “Victor Babes” University of Medicine and Pharmacy, 300041 Timisoara, Romania; raluca.dumache@umft.ro; 3Department of Surgery, “Victor Babes” University of Medicine and Pharmacy, 300041 Timisoara, Romania; ciprian_duta@yahoo.com; 4Department of Pathology, Victor Babes” University of Medicine and Pharmacy, 300041 Timisoara, Romania; vita.octavia@umft.ro; 5Faculty of Automation and Computer Science, Politehnica University, 300223 Timisoara, Romania; marina.mercioni@cs.upt.ro; 6Department of Obstetrics and Gynecology, “Victor Babes” University of Medicine and Pharmacy, 300041 Timisoara, Romania; 7Advanced Regional Research Center in Gastroenterology and Hepatology, “Victor Babes” University of Medicine and Pharmacy, 300041 Timisoara, Romania; sirli.roxana@umft.ro; 8Cardiology Department, “Victor Babes” University of Medicine and Pharmacy, 300041 Timisoara, Romania; iurciuc.stela@umft.ro

**Keywords:** *SMAD7*, colorectal cancer, disease-free survival, polymorphisms, single nucleotide polymorphism

## Abstract

Colorectal cancer (CRC) is globally recognized as a prevalent malignancy known for its significant mortality rate. Recent years have witnessed a rising incidence trend in colorectal cancer, emphasizing the necessity for early diagnosis. Our study focused on examining the impact of the *SMAD7* gene variant rs4939827 on the risk of colorectal cancer occurrence. The composition of our study group included 340 individuals, comprising 170 CRC diagnosed patients and 170 healthy controls. We performed genotyping of all biological samples using the TaqMan assay on the ABI 7500 Real-Time PCR System (Applied Biosystems, Waltham, MA, USA). This investigation focused on the rs4939827 gene variant, assessing its association with CRC risk and clinicopathological characteristics. Genotyping results for the *SMAD7* gene variant rs4939827 revealed a 42.6% prevalence of the C allele in CRC patients (*p* = 0.245) and a 22.8% prevalence of the T allele in control subjects (*p* = 0.109). This study concluded that there was an elevated risk of CRC in the dominant model for CC/CT+TT, with a *p*-value of 0.113 and an odds ratio (OR) of 2.781, within a 95% confidence interval (CI) of 0.998 to 3.456. The findings of our research indicate a correlation between variants of the *SMAD7* gene and the likelihood of developing colorectal cancer in our study population. Consequently, these results could be instrumental in facilitating early diagnosis of colorectal cancer utilizing information on single-nucleotide polymorphism (SNP) and genetic susceptibility to the disease.

## 1. Introduction

Breast cancer is the most often diagnosed cancer in the EU, accounting for an estimated 380,000 cases (99% of which are females), and accounting for 13.8% of all cancer diagnoses. Following this are colorectal cancer (356,000; 13% of all new cases), prostate cancer (330,000; 12.1%), and lung cancer (319,000; 11.6%). Lung cancer is predicted to account for 19.5% of all cancer fatalities in the EU, followed by colorectal (12.3%), breast (7.5%), and pancreatic cancer (7.4%) [1,2].

According to the World Health Organization (WHO), it is projected that by 2040, the number of new instances of colorectal cancer will rise to 3.2 million per year, which represents a 63% increase. Additionally, the number of fatalities caused by colorectal cancer is expected to reach 1.6 million per year, reflecting a 73% increase [3].

Incidence rates are approximately four-fold higher in transitioned countries compared with transitioning countries, but there is less variation in the mortality rates because of higher fatality in transitioning countries [4]. In Europe, colorectal cancer is the most common type of cancer, with about 450,000 people newly diagnosed each year [5]. In Romania in 2020, this type of cancer was the second-most common. Compared to other European countries, Romania is ranked fourth for mortality in men and women due to CRC [6]. This neoplasm originates from various precursor lesions, such as conventional adenomas and serrated polyps. These diverse origins contribute to the evolution into carcinoma via different pathways. The majority of CRC cases follow the adenoma–carcinoma sequence. However, serrated polyps, formerly referred to as hyperplastic polyps, are also recognized as precursor lesions leading to an alternate CRC development pathway. Furthermore, the likelihood of CRC development is affected by environmental and genetic factors. A portion of this inherited predisposition to CRC has been identified, encompassing both infrequent, high-penetrance and frequent, low-penetrance genetic variations [7].

Colorectal cancer is defined as a cancerous growth which can occur in the colon, rectum, or appendix. It includes a large spectrum of neoplasia, which can range from precancerous to invasive cancers and are generally epithelial-derived tumors (e.g., adenocarcinomas or adenomas) [8,9].

CRC can be divided into familial CRC (hereditary CRC) and sporadic CRC (non-hereditary CRC). It has been demonstrated that more than 35% of the variation in CRC susceptibility depends on genetic factors and on genetic mutations [10,11,12]. Hereditary CRC represents about 25% of all CRCs and is linked to high-penetrance mutations [11,13].

It has been considered that CRC is caused by the interactions between environmental factors and genetic variants [14,15]. Genetic studies have revealed that non-hereditary CRC is caused by genetic defects known as single-nucleotide polymorphisms (SNP), rather than genetic mutations. Thus, understanding which genetic variants occur during the carcinogenesis of non-hereditary CRC is important in the prevention, screening, and early diagnosis of CRC [16,17,18], which, together with colorectal cancer screening, is known to decrease mortality [19]. 

Currently, the guidelines recommend screening for colorectal cancer using fecal occult blood testing, sigmoidoscopy, or colonoscopy in adults, beginning at age 50 and continuing until age 75 years, depending on the local socio-economic conditions and availability of different screening tests. CT colonography and FIT-DNA testing are acceptable “*alternative tests*” [20]. 

Considering the rising incidence of early CRC (<50 years old) in the United States, American guidelines recommend an earlier start of the screening programs, namely at 45, mainly using colonoscopy and fecal immunochemical testing (FIT), or using CT colonography, colon capsule, flexible sigmoidoscopy, and multitarget stool DNA testing in patients unwilling or unable to undergo FIT or colonoscopy [21].

A single-nucleotide polymorphism (SNP) represents a type of genetic variation that can result in different outcomes, influencing an individual’s vulnerability to diseases. It serves as a potential biomarker for estimating cancer risks, including that of CRC [22,23]. Over the past three decades, genome-wide association studies (GWAS) have identified numerous genetic loci linked to CRC risk. Among these, some are part of the TGF-β signaling pathway. Notably, one prevalent polymorphism associated with increased CRC risk is found in the *SMAD7* gene [24]. Tenesa and colleagues pinpointed a specific genetic variant, rs4939827, located on chromosome 18q21 within *SMAD7*. This variant is linked to CRC risk with an odds ratio (OR) of 1.2 and a *p*-value of 7.80 × 10^−28^ [25]. Worldwide screening programs for diagnosing CRC in its early stages are improving and provide the possibility of early diagnosis along with personalized therapy [26]. 

The *SMAD7* gene is involved in inflammation-related pathways and modulates transforming growth factor (TGF)-β and Wnt signaling, both of which are known to be involved in the development of colon tumors [27]. 

In this research, a case–control study was conducted to evaluate the association between the *SMAD7* gene variant rs4939827 and the risk of colorectal cancer in the western Romanian population.

## 2. Materials and Methods

### 2.1. Study Population Characteristics

*SMAD7* rs4939827 was associated with a higher risk of colorectal cancer in both the recessive and co-dominant models, according to the study literature, which served as a technique for choosing the genetic markers and the importance of the haplotype analysis.

Our research involved 170 CRC patients and 170 control patients with non-colonic conditions. All CRC cases were confirmed through positive colonoscopy results and pathology reports indicating malignant tumors in the colon or rectum. These patients were enrolled from 1 June 2020 to 1 June 2022 through the cancer registry at the Gastroenterology and Hepatology Clinic of the “Pius Brinzeu” Emergency Hospital in Timisoara, Romania. Informed consent was obtained from all participants before the study. Clinical data, pathological grades, and clinical stages were gathered from medical records. Histological classification and pathological staging adhered to the UICC’s TNM classification (8th edition, 2020) for malignant tumors in the colon and rectum [28]. The control group comprised individuals with normal colonoscopy results and no familial history of gastrointestinal diseases. This study received approval from the Ethics Committee of the “Victor Babes” University of Medicine and Pharmacy in Timisoara, Romania, under protocol no.27/25.06.2020. Colorectal cases were classified based on the AJCC’s 8th edition criteria [29].

A total of 170 patients with CRC were included: 65 (38.23%) females and 105 (61.76%) males, as well as 170 controls. The mean age of the CRC patients was 59.57 ± 28.09 years and that of the controls was 61.36 ± 23.10 years.

In Table 1, the demographic characteristics and histological parameters of the study population are presented.

### 2.2. DNA Isolation and Genotyping

After inclusion in the study, 3 mL blood samples were collected from all the participants in EDTA tubes. Further, DNA was isolated from the biological samples using the Maxwell RSC Whole Blood DNA kit (Promega, Madison, WI, USA). The isolation was performed with the Automate DNA/RNA Maxwell RSC 48 System (Promega, USA). Genotyping of *SMAD7* was performed with a 7500 real-time PCR equipment.

In all cases, rs4939827 was genotyped using the TaqMan SNP Genotyping Assay (ThermoFisher Scientific, Waltham, MA, USA). All the DNA samples were genotyped with a 7500 Real-time PCR equipment (Applied Biosystem, USA) in a 25 µL reaction volume containing TaqMan Genotyping Assay and TaqMan PCR Master Mix and the DNA sample. The genotypes were analyzed by measuring allele-specific fluorescence using software for allelic discrimination (ThermoFisher Scientific, USA).

### 2.3. Statistical Analysis

The statistical analyses in our study were performed using SPSS software, version 28. We compared allele and genotype frequencies, along with clinicopathological features, employing the χ2 test. To adjust for confounding factors like age and gender, linear regression analysis was utilized to compute the odds ratio (OR) and its 95% confidence intervals (95% CI). The χ2 test was also used to identify differences in demographic factors. Data were deemed significant when *p* values were below 0.05 in all comparisons. Additionally, in this study, the allele with the lowest frequency was designated as the “risk allele.” For this study, the dominant model was used to assign 0 (if the risk allele was not present) and 1 (if one risk allele was present), and the recessive model assigned 0 (if the risk allele was not present) and 1 (if two copies of the risk allele were present).

## 3. Results

Results were considered statistically significant for a *p* value ≤ 0.05. From the statistical analyses of rs4939827, χ2 = 1.53 and *p* = 0.57 (*p* < 0.05) were obtained. In our study, the variant respected the Hardy–Weinberg equilibrium. The prevalence of each of the rs493987 genotypes, CC, CT, and TT, was as follows: 42.6%, 39.7%, and 46.5%. The most frequent allele was found to be the C allele. Table 2 presents the frequencies of the CRC risk alleles in the study group, from which it can be observed that the CT genotype is quite balanced, with 57 (0) and 45 (1), accounting for 34.1% and 26.9% of the data, which are the highest percentages. 

In Table 3, the regression results, based on a linear model (estimated using ML) to predict colorectal cancer (CRC) associated with gender, age, and genotypes (formula: colorectal cancer ~ (gender + age + genotypes) − age), are presented. The model’s explanatory power is substantial (R2 = 0.27). Gender F-F has the biggest standard error (SE) at 0.7672, while age has the lowest at 0.0178. The standard error often decreases as the number of data points included in the mean computation increases. The number of standard deviations by which a value deviates from the mean of a particular distribution is shown by the z-score (Z). Z-scores that are negative show that the value is below the mean. Z-scores that are positive show that the value is above the mean (age, genotype CC–CT). The likelihood of receiving a result that is as severe as or more extreme than what was observed, under the premise that there is no impact or difference (null hypothesis), is known as the *p* value. *p*, or probability, expresses the likelihood that any observed variation across groups is the result of random variation. All *p* values are greater than the 0.05 threshold, which suggests no statistical significance. Utilizing the odds ratio (OR), one may ascertain if a certain exposure contributes to a given result and evaluate the relative importance of other risk variables for that same event. The OR values can be interpreted as follows:OR = 1. Odds of outcome are unaffected by exposure (age).Exposure is linked to increased outcome odds (OR > 1) (genotype CC–CT).Exposure at OR < 1 is linked to decreased outcome chances.
diagnostics-14-00220-t003_Table 3Table 3Model coefficients—CRC.
95% CIPredictorEstimateSEZ*p*ORLowerUpperIntercept1.402831.16751.20160.234.06670.412540.089Age0.00460.01780.25920.7961.00460.97021.04Genotype






TT–CC−0.347510.4581−0.75860.4480.70640.28781.734CC–CT0.554940.59230.93690.11392.780.54555.561Gender






F–F−1.359780.7672−1.77250.0760.25670.05711.155M–F−0.326880.3979−0.82160.4110.72120.33061.573**Abbreviations**: OR = odds ratio; CI = confidence interval.

As it is presented in Table 2, CT heterozygous carriers have the highest risk of developing CRC in the studied group (26.9%), followed by CC homozygous carriers and TT homozygous carriers.

According to the data presented in Table 3, individuals with heterozygous carriage of CT exhibit an elevated susceptibility to colorectal cancer (CRC) in comparison to homozygous carriers. This heightened risk is substantiated by the observation that their odds ratio (OR) value attains the highest magnitude among the studied groups.

### Association between Colorectal Cancer (CRC) Risk and Genetic Polymorphisms

The results of association between the *SMAD7* genotypes and clinicopathological characteristics in CRC patients are shown in Table 4. 

Figure 1 and Figure 2 present the microscopic aspect of colon adenocarcinoma in stage III.

Microscopic investigations have shown distinct histological characteristics that are unique to these particular forms of tumors. Certain tubular adenocarcinomas have regions with a substantial presence of mucus, resembling both mucinous adenocarcinomas and signet-ring tumor cells. The presence of these microscopic features serves as evidence for the significant phenotypical diversity shown by the cells comprising the architecture of a colorectal tumor.

## 4. Discussion

One of the most important problems in CRC is its late diagnosis; therefore, there is an urgent need for molecular studies which can aid in early diagnosis of this disease [30]. 

In our study, a significant association between the *SMAD7* gene variant rs4939827 (C > T) on chromosome 18q21.1 and an increased risk of CRC was found in the western Romanian population in both studied groups. The presence of the C allele in the colorectal cancer group represents a predictive risk factor for the development of the disease. 

Studies in the field of molecular genetics have demonstrated that molecular variations which are related to some disorders may aid in the preventive strategy against CRC in the general population. SMADs act as receptors and signaling transducers which are involved in the TGF-β signaling pathway. *SMAD7* can be binned as a TGF-β receptor type I and thus acts as a negative regulator of the TGF-β signaling pathway. 

Yan X and colleagues [26] have shown that the TGF-β signaling pathway is impaired due to *SMAD7*, leading to an increased risk of colorectal cancer. Several studies have established a notable link between the rs4939827 variant and a heightened risk of CRC [31,32,33,34]. Broderick et al. [35] conducted a genome-wide association study focusing on the *SMAD7* gene on chromosome 18q21. They discovered extensive polymorphism in intron 3 of *SMAD7*, particularly on the SNP rs4939827. Additionally, Boulay et al. [31] investigated the clinical relevance of a deletion in *SMAD7*. They concluded that *SMAD7* could act as a prognostic indicator in CRC patients and plays a significant role in tumor suppression. Thus, the loss of *SMAD7* may render one more susceptible to carcinoma cells.

A study led by Phipps et al. [36] explored the connection between the rs4939827 variant and the survival of 2611 CRC patients. Their research suggested that this variant could influence CRC progression [29,37].

Similarly, our findings align with those presented by Xiong et al. [38]. In their study, they identified five single-nucleotide polymorphisms (SNPs)—rs6983267, rs4939827, rs10795668, rs3802842, and rs961253—that showed a statistically significant correlation with the risk of colorectal cancer. Their research also indicated that individuals carrying four or more of these risk genotypes had a 3.25 times greater risk of developing colorectal cancer compared to those without any of these risk genotypes. The confidence interval for this finding was 95%, ranging from 2.24 to 4.72 [38].

Similar to our findings, Ho et al. [39] showed that out of 13 single-nucleotide polymorphisms that were genotyped during Phase 1 of their study, a significant association with CRC risk was observed for 5 of them (rs10795668, rs7014346, rs12953717, rs4779584, and rs4939827). According to the authors, the risk-increasing allele for all five single-nucleotide polymorphisms examined corresponds to the allele reported in the original association study. The two aforementioned single-nucleotide polymorphisms, namely rs4939827 and rs12953717, located on chromosome 18q21.2, exhibit a robust linkage disequilibrium with one another [39].

Further research in the literature reveals that Alidoust et al. [40] found a statistically significant correlation between the rs4939827 single-nucleotide polymorphism and an elevated risk of colorectal cancer [*p*  =  0.02, odds ratio (OR)  =  2.49, 95% confidence interval (CI) (1.13–4.49)]. This result agrees with our conclusions, showing that the SNP under investigation shows an association with colorectal cancer (CRC) risk in both the recessive and co-dominant genetic models. Specifically, individuals with the TT/CT+CC genotype had a higher odds ratio (OR) of 2.90, with a 95% confidence interval (CI) ranging from 1.38 to 6.09, and a *p*-value of 0.005. Similarly, those with the CC+TT/CT genotype had an OR of 1.66, with a 95% CI ranging from 1.00 to 2.75, and a *p*-value of 0.01 [40].

In the study in question, haplotype analysis identified two haplotypes linked to an increased risk of colorectal cancer (CRC) [36]. These are the T-C haplotype of rs8085824-rs34007497 and the T-C and C-T haplotypes of rs8085824-rs4939827. Conversely, the C-C haplotype of rs8085824-rs4939827 was associated with a lower risk of developing CRC. Additionally, the frequencies of three haplotypes—C-C-T, C-C-C, and T-C-C—involving rs8085824, rs34007497, and rs4939827 were found to differ between CRC patients and healthy individuals. Moreover, the A-T-C and A-C-T haplotypes, incorporating the genetic variants rs8088297, rs8085824, and rs4939827, were linked to an increased risk of colorectal cancer [40].

A meta-analysis of 63 studies conducted by Huang et al. [41] revealed that the C allele of rs4464148 (CC vs. TT+TC, [OR] = 1.23, 95% confidence interval [CI]: 1.14–1.33, *p* < 0.01), the T allele of rs4939827 (TT vs. CC+TC, OR = 1.15, 95%CI: 1.07–1.22, *p* < 0.01), and the T allele of rs12953717 (TT vs. CC+TC, OR = 1.22, 95%CI: 1.16–1.29, *p* < 0.01) were all found to be significantly associated with an increased risk of colorectal cancer.

The results of subgroup analysis based on ethnicity indicate a significant association between rs4939837 and colorectal cancer risk in the Caucasian population [37]. This association was observed in 27 out of the 63 studies. The comparison of the TT genotype with the TC+CC genotypes yielded an odds ratio of 1.19 ([CI]: 1.12–1.26), with a *p*-value less than 0.01. Heterogeneity analysis showed no significant variation among the studies (*p*-value for heterogeneity [PH] = 0.00). On the other hand, nine reports analyzed in the same meta-analysis show that no association between rs4939837 and CRC risk was found in the Asian population [41]. 

Another analysis containing a total of 34 studies looked at rs4939827 minor allele (C) [42]. It has been shown to be a protective factor against colorectal cancer, thus endorsing our results. The odds ratios and corresponding 95% confidence intervals for several genetic models in this study were as follows: dominant model, OR = 0.89 (95% CI: 0.83–0.97); recessive model, OR = 0.89 (95% CI: 0.83–0.96); homozygous model, OR = 0.84 (95% CI: 0.76–0.93); heterozygous model, OR = 0.91 (95% CI: 0.85–0.97); additive model, OR = 0.91 (95% CI: 0.87–0.96). It may be assumed that the presence of the rs4939827 variant (T > C) may potentially reduce susceptibility to CRC. The SNPs rs4939827, rs4464148, and rs12953717 have been extensively investigated in relation to many types of cancer. Among the several options, CRC stands out as the most prominent [42].

As previously indicated, a diverse range of polymorphisms located at various places on the *SMAD7* genes were chosen in order to provide a comprehensive understanding of the whole gene [40]. One notable aspect of this work is in its comprehensive examination of several markers using gene and haplotype analysis, hence enhancing the statistical power of the investigation [43]. To the extent of our current understanding, none of the preceding investigations have examined the haplotypes of this gene in relation to their connection with colorectal cancer. Furthermore, the confirmation of the link between the rs4939827 variant and colorectal cancer within our study group might enhance the substantiation of the impact of this genetic risk factor on CRC. This finding has potential significance in the realm of clinical risk evaluation, particularly in the context of direct-to-consumer genetic testing [30].

As a limitation, the chosen genetic markers might not capture the full genetic complexity of colorectal cancer. To mitigate this, it is considered necessary for future research to explore additional markers.

Although some researchers explain alternate racial/ethnic classifications based on disparities in experiences with healthcare across racial/ethnic minority groups, this study had an interest in the experiences of all self-identified individuals, regardless of race or ethnicity [44].

## 5. Conclusions

To the best of our understanding, this is the inaugural case–control study examining the impact of the rs4939827 variant on clinical characteristics and the risk of CRC within the Romanian population. Our findings indicate a significant association between the rs4939827 variant and an increased risk of CRC in this demographic.

The findings of our study indicate a potential association between the rs4939827 variant and susceptibility to colorectal cancer in the Romanian population. Additional large-scale population-based research is required in the future to validate the precision of the results and investigate potential gene–environment interactions. In order to obtain a comprehensive understanding of the fundamental processes behind CRC formation, it is essential to conduct future research that investigates the functional activity of these genetic variants and their impact on carcinogenesis.

Among the various biomarkers that can be detected from biological samples, rs4939827 displays good diagnostic performance for prediction of high-risk colon cancer, which would make this biomarker a promising testing tool for early detection.

In summary, we conclude that using this biomarker in CRC screening programs could aid in earlier prediction of CRC risk in the Romanian population.

## Figures and Tables

**Figure 1 diagnostics-14-00220-f001:**
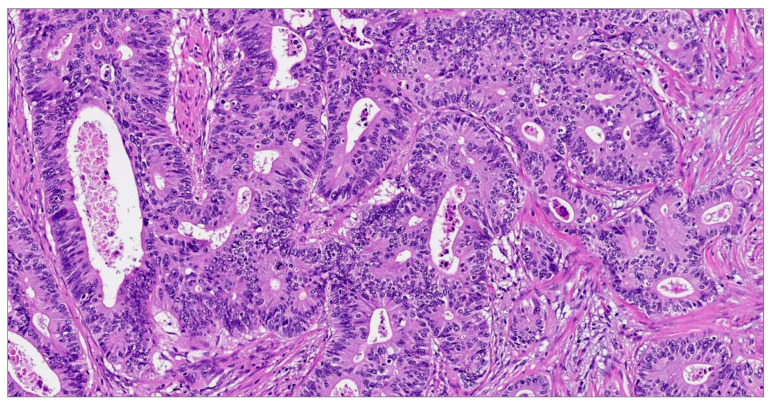
Moderately differentiated conventional adenocarcinomas (the usual type) with complex, irregular, crowded glands and cribriform aggregates. The tumor invades the muscularis propria. Hematoxylin–eosin stain ×100.

**Figure 2 diagnostics-14-00220-f002:**
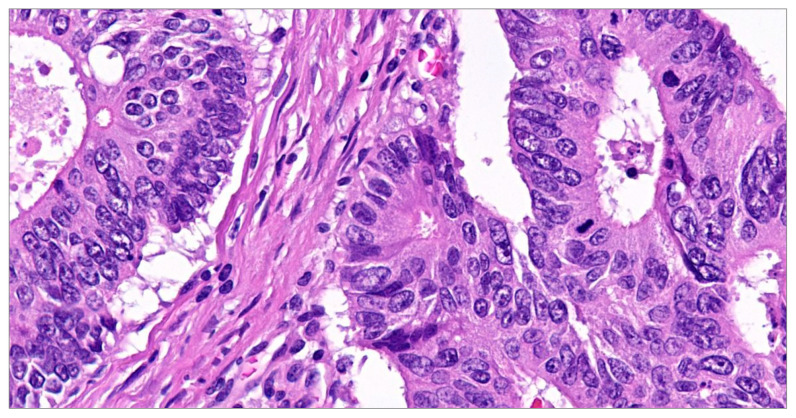
Moderately differentiated conventional adenocarcinomas (the usual type)—loss of nuclear polarity, nuclear pleomorphism, mitotic activity present. Hematoxylin–eosin stain ×400.

**Table 1 diagnostics-14-00220-t001:** Demographic characteristics of the patients and controls and tumor characteristics.

Parameters	Colorectal Cancer Cases	Controls
**Number**	170	170
**Sex**		
Male	105 (61.76%)	85 (50%)
Female	65 (38.23%)	85 (50%)
**Age (years)**	59.57 + 28.09	61.36 + 23.10
**AJCC stage**		
I	86	-
II	35	-
III	45	-
IV	4	-
**Tumor Size (cm)**	4.5 + 12.8	-
**Location**		
colon	132 (77.64%)	-
rectum	27 (15.88%)	-
rectosigmoid	11 (6.48%)	-

**Abbreviations**: cm = centimeters.

**Table 2 diagnostics-14-00220-t002:** Frequencies of homozygote and heterozygote cases in CRC patients: CT is associated with a 26.9% risk of CRC, followed by CC with 10.8% and TT with 8.4%.

Genotype	CRC	Counts	% of Total	Cumulative %
CC	0	21	12.6%	12.6%
	1	18	10.8%	23.4%
CT	0	57	34.1%	57.5%
	1	45	26.9%	84.4%
TT	0	12	7.2%	91.6%
	1	14	8.4%	100.0%

**Table 4 diagnostics-14-00220-t004:** The frequencies of the *SMAD7* gene single-nucleotide polymorphism in 170 patients with colorectal cancer.

Gene	rs Chr Position GR Ch38	AA/AB/BB	AA%	AB%	BB%
SAMD7	rs 4939827	CC/CT/TT	69 (42.6%)	77 (39.7%)	24 (17.7%)

**Abbreviations**: gene names as per international nomenclature HUGO Gene Nomenclature Committee (HGNC); Chr = chromosome position reference genome GRCh; AA = homozygous for one allele; AB = heterozygous; BB = homozygous for the other allele; SAMD7 = sterile alpha motif domain containing 7.

## Data Availability

Data available on request.

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
