# Peer review of "Colorectal Cancer Risk Prediction Using the rs4939827 Polymorphism of the SMAD7 Gene in the Romanian Population"

_diagnostics, 2024, doi:10.3390/diagnostics14020220_

Round 1

Reviewer 1 Report

Comments and Suggestions for Authors

Investigation of the SMAD7 gene variant and its potential association with colorectal cancer (CRC) in the Romanian population. The author's study's comprehensive approach, including genotyping with TaqMan assay and statistical analyses, contributes valuable insights into the relationship between rs4939827 polymorphism and CRC risk. The inclusion of demographic characteristics and histological parameters enhances the study's robustness. The observed increased risk of CRC in the dominant model for CC/CT+TT and the association with the C allele in the colorectal cancer group are noteworthy findings. The integration of clinicopathological characteristics further strengthens the potential clinical relevance of your results. The extensive discussion, citing various studies and meta-analyses, provides a broader context for your findings. T

I recommend addressing the following points:

1. Clarify the interpretation of the results related to the frequencies of rs493987 genotypes and their association with CRC risk in the study group.

2. Provide additional details on the methodology, particularly the rationale for selecting the genetic markers and the significance of the haplotype analysis.

3. Consider discussing potential limitations and challenges in your study, such as sample size or any biases that might impact the generalizability of the results. 

Author Response

Thank you for your feedback. Please see the attachment "author coverletter v1.docx"

Please ignore the author-coverletter-33711164.v1 docx

Reviewer 2 Report

Comments and Suggestions for Authors

Comments to the Authors

This is a review of the paper entitled “Risk Prediction of Colorectal Cancer using rs4939827 polymorphism of SMAD7 gene in Romanian Population”, by Herlo et al.

Dear authors,

This work is focused on a very relevant field and the content is interesting… Nonetheless, I have some reservations regarding the paper. Since it is innovative, I propose acceptance after minor revisions. See my concerns below:

Introduction

Line 42 – 44: Why used data from 2020 if you can easily consult updated data for 2023 (maybe even some prevision to 2024) of these numbers. WHO, for example, usually provides this type of estimations.

Line 48: Again data from almost 4 years ago… I am certain that the authors can easily update this portion with newer data.

Line 62: familiar…

The entire paragraph between line 62 and 76 is very confusing and addressed very distinct topics creating a mess of a paragraph hard to understand. I would recommend the authors to organize the paragraph considerably better and even split it in several paragraphs dedicated to all the topics here addressed. For example, the authors start by describing the types of CRC, but then jump to the importance of screening and diagnosis, and jump again to the different types of employed techniques… The paper would benefit of more organization.

Line 87: … have found that of the most… English improvements are required throughout the entire document. This is just one of many examples.

As I mentioned, the introduction is, in overall, a bit confusing and hard to follow. It is rich in jumps from one topic to another, and then jumping back to the same topic… Please reformulate the writing haven per base the typical cause-consequence approach usually employed in scientifical writing. In addition, the paper would benefit of a bit more of information. This is an important topic that deserves to be better introduced and addressed.

Material and Methods

Congratulations on gathering such an elevated number of volunteers. I know how hard is to create a good cohort of volunteers.

In addition, the authors should have included some demography regarding the cohort… gender, age, and so on…

Line 122 and others: the authors often use the first person to write… I would recommend the adoption of a more scientific writing, i.e., using the third person (the cohort was.. the samples were collected…)

Line 129: how do you calculate statistical analysis? You can perform statistical analysis or calculate some values… not calculate statistical analysis.

Statistical analysis: I find the approach interesting but it is rather har to follow. I recommend the authors to provide more information on the approach. I understand that they used linear regressions to adjust confounding variables, and I’m ok with it… but further details on the calculation of odds ratio should be provided. Again, I am comfortable with the use of χ2 test by the authors, but I would like to read more details on the assumptions (0 and 1) used for both dominant and recessive models…

Results

It is the first time that I see the characteristics of the population addressed in the results chapter… I would move it to the chapter 2.1, where it belongs.

Table 1 has some cells not properly formatted.

Table 3: since the authors included the meaning of the abbreviations OR an CI, then they should have also included the meaning of SE, Z, …

The authors did not include any kind of discussion regarding table 2 and 3… even if they are “only“ statistical data, they are results that must be discussed.

I know that the authors discussed the results on a separate section, however, this makes the paper less interesting… in my point of view, each portion of the results should be followed by the respective discussion… otherwise, the results, which are very interesting, lose some of the logic…

The overal paper has a total of 41% of plagiarism according to turnitin, please fix it...

The paper is interesting and innovative. I propose acceptance after minor revisions.

Comments on the Quality of English Language

Moderate editing of English language required

Author Response

Thank you for your feedback. Please see the attachment below.

Reviewer 3 Report

Comments and Suggestions for Authors

It is an interesting and well-written study. It would be of added value if you could also provide the MSI status of the examined patients and see if there is a correlation between the examined SNP and MSI high status

Author Response

Thank you for your feedback. Please see the attachment.
